# Correlation of Morphology and In-Vitro Degradation Behavior of Spray Pyrolyzed Bioactive Glasses

**DOI:** 10.3390/ma12223703

**Published:** 2019-11-09

**Authors:** Fetene Fufa Bakare, Yu-Jen Chou, Yu-Hsuan Huang, Abadi Hadush Tesfay, Toshihiro Moriga, Shao-Ju Shih

**Affiliations:** 1Department of Materials Science and Engineering, National Taiwan University of Science and Technology, Taipei 10607, Taiwan; d10504825@mail.ntust.edu.tw (F.F.B.); M10704322@mail.ntust.edu.tw (Y.-H.H.); d10504802@mail.ntust.edu.tw (A.H.T.); 2Department of Mechanical Engineering, National Taiwan University of Science and Technology, Taipei 10607, Taiwan; yu-jen.chou@mail.ntust.edu.tw; 3Department of Chemical Science and Technology, Graduate School of Advanced Technology and Science, Tokushima University, 2-1 Minami-Josanjima, Tokushima 770-8506, Japan; moriga@chem.tokushima-u.ac.jp

**Keywords:** bioactive glass, PEG, electron microscopy, morphology, in-vitro degradation

## Abstract

Bioactive glass (BG) is considered to be one of the most remarkable materials in the field of bone tissue regeneration due to its superior bioactivity. In this study, both un-treated and polyethylene glycols (PEG)-treated BG particles were prepared using a spray pyrolysis process to study the correlation between particle morphology and degradation behavior. The phase compositions, surface morphologies, inner structures, and specific surface areas of all BG specimens were examined by X-ray diffraction, scanning electron microscopy, transmission electron microscopy, and nitrogen adsorption/desorption, respectively. Simulated body fluid (SBF) immersion evaluated the assessments of bioactivity and degradation behavior. The results demonstrate three particle morphologies of solid, porous, and hollow factors. The correlation between porosity, bioactivity, and degradation behavior was discussed.

## 1. Introduction

The superior bioactivity of bioactive glass (BG) has been demonstrated to bond with living tissues and to stimulate new tissue growth due to their similar compositions in the main inorganic components of human bones [1,2]. In addition, properties such as biocompatibility and degradability have also received numerous attention in the fields of tooth fillers, drug carriers, and bone implants in recent years [3,4,5]. Among these properties, Fiume et al. provided a general review with their use in clinics in recent decades [6], where the use of porous scaffolds and the degradation behavior becomes more important due to its influences on migration and proliferation of new tissue formation as well as the extended technological progress of manufacturing and functionalization. Therefore, the BG’s degradation ability is critical for future tissue engineering development.

According to the previous studies [7,8,9,10], two common factors of composition and porosity have been used to manipulate the degradation behavior of BG. Initially, for composition, Hill correlated the numbers of network connectivity (number of bridging oxygen per silicon-oxide tetrahedron [7]) to the solubility and degradation behaviors of BG. The study suggested that lower network connectivity will lead to a greater degradation rate (or higher solubility) and better bioactivity [8]. Then, the factor of porosity has been demonstrated by Zhang et al. showing that scaffolds with a porous structure have a rapid degradation rate along with better bioactivity for bone regeneration applications [9]. Deliormanh prepared the BG scaffolds with the filament diameters of 130 and 300 μm and found that the scaffolds with the thinner diameters have a higher degradation rate [10]. Although both composition and porosity can influence BG’s degradation properties, the effects of composition on degradation is well established. Thus, this study attempts to control the degradation behavior of BG by controlling the porosity factor.

So far, three methods of sol-gel, spray drying, and spray pyrolysis (SP), have been commonly used for synthesis of BG powders. The sol–gel method is a versatile process that offers low calcination temperature and chemical flexibility [11]. However, the whole process is a batch production that takes several days. It is difficult to control the morphology [12,13]. Next, the spray drying method gives high control over shape and morphology through a fast kinetic process. However, when the process is performed by spraying inflammable solvents, both safety and economic concerns will rise in view of an effective process [14,15]. Instead, the SP method offers the following merits: easy to operate, economical, continuous processing, and short process time [16,17]. In addition, Polyethylene glycols (PEG), which are common pore forming agents [18], are proposed as an additive in this study. It is a highly hydrophilic polymer, which is soluble in BG precursor’s solution including water and ethanol solvent [19]. Furthermore, its decomposition temperature (450 °C) [20] is much lower than the calcination temperature of BG (550 °C) [21], which allows the decomposition reaction of PEG to complete.

It is well known that porosity dominates the degradation behavior of BG [22]. Therefore, in this study, we aim to control the porosity of spray pyrolyzed BG particles using various PEG concentrations and investigate the relationship between porosity, bioactivity, and degradation ability. To begin with, the un-treated BG particles and the 0.1 M, 0.3 M, and 0.5 M PEG-treated BG particles were prepared using the SP process. Moreover, phase compositions, surface morphologies, inner structures, and specific surface area were observed using X-ray diffraction (XRD), scanning electron microscopy (SEM), transmission electron microscopy (TEM) and the nitrogen adsorption/desorption method (Brunauer-Emmett-Teller (BET) method), respectively. In addition, the bioactive tests were carried out and characterized using a Fourier transform infrared spectroscopy (FTIR). At last, the degradation behaviors of all BG specimens were examined and the formation mechanism of porosity was discussed.

## 2. Materials and Methods

### 2.1. Synthesis

First, the un-treated BG particles were prepared by SP using the common composition of 58S (60 mol % SiO_2_, 36 mol % CaO, and 4 mol % P_2_O_5_). The precursor solution was prepared by mixing 37.49 g tetraethyl orthosilicate (TEOS, Si(OC_2_H_5_)_4_, 99.9 wt %, Showa, Osaka, Japan), 25.50 g calcium nitrate tetrahydrate (CN, Ca(NO_3_)_2_·4H_2_O, 98.5 wt %, Showa, Osaka, Japan), and 4.37 g triethyl phosphate (TEP, (C_2_H_5_)_3_PO_4_, 99 wt %, Alfa Aesar, Haverhill, MA, USA) into 60.00 g of ethanol with 1.60 g of 0.5 M HCl. For PEG-treated BG particles, the precursor solutions were prepared by adding additional polyethylene glycol (PEG, 95.0 wt %, molecular weight of 600g/mol, Showa, Tokyo, Japan) with various concentrations of 0.1 M, 0.3 M, or 0.5 M. Then, all solutions were stirred at room temperature for 2 h to achieve homogeneity. Next, for the SP process, the precursor solutions were poured into an ultrasonic atomizer (KT-100A, King Ultrasonic, New Taipei, Taiwan) operating at the frequency of 1.65 MHz. The droplets were generated and fed into a tube furnace (D-80, Dengyng Co., New Taipei, Taiwan) with three heating zones, which were set at 250 °C, 550 °C, and 350 °C, corresponding to the evaporation, calcination, and cooling process of SP. Lastly, the resulting particles were charged at the voltage of 16 kV and collected in an earthed stainless steel collector.

### 2.2. Characterization

The phase compositions of un-treated, 0.1 M, 0.3 M, and 0.5 M PEG-treated BG powders were characterized by using the X-ray diffractometer (D2 Phaser, Bruker, Karlsruhe, Germany) with the Cu-Kα radiation. These diffraction patterns were collected with a diffraction angle ranging from 20° to 80° and a scanning rate of 6° per minute. Both surface morphology and the inner structure are required to obtain the three-dimensional morphology [23]. Therefore, a field-emission scanning electron microscope (JSM-6500F, JEOL, Tokyo, Japan) and a field-emission transmission electron microscope (Tecnai G2 F20, FEI, Hillsboro, OR, USA) were employed, while the particle sizes were calculated from more than 300 particles from several SEM images. It should also be noted that the particle size is mainly attributed from the factors of precursor concentration, ultrasonic frequency, and precursor composition [24], and, in this study, only the factor of the pore forming agent of PEG was altered since the same ultrasonic frequency and the precursor compositions were used. Therefore, the porosity of each PEG-treated BG particles can be estimated using the average particle sizes of un-treated and PEG-treated BG powder with Equation (1) below.

Porosity (%) = (V_2_ − V_1_)/V_2_ × 100%
(1)
where V_1_ is the volume of the solid particles (un-treated particles), and V_2_ is the volume of the porous or hollow particles (PEG-treated particles).

In addition, the specific surface areas were measured by BET (Brunauer–Emmett–Teller). The nitrogen adsorption and desorption isotherms were operated at −196 °C on a constant-volume adsorption apparatus (Novatouch LX2, Quanta chrome Instruments, Boynton Beach, FL, USA). All specimens were degassed at 200 °C for 3 h before the measurements to ensure a fair comparison.

The bioactivity of all BG specimens was examined by immersing BGs in simulated body fluid (SBF), which has an ionic concentration similar to human plasma [25]. The amount of SBF was adjusted based on the specific surface area of specimens [26]. Thus, the solid to liquid ratios were 2 mg to 10 mL for the un-treated BG powder, 2 mg to 14 mL for the 0.1 M PEG-treated BG powder, 2 mg to 18 mL for the 0.3 M PEG-treated BG powder, and 2 mg to 22 mL for the 0.5 M PEG-treated BG powder, respectively. Then, each specimen was immersed in SBF and water-bathed at the constant temperature of 37 °C for one day. After removing SBF, the samples were washed by acetone and deionized water and dried in an oven overnight. The bioactivity was examined using SEM, XRD, and FTIR measurements, while the bioactivity is determined by obtaining the FTIR peak intensity of the P-O bending vibration around 566 cm^−1^ [2] and the intensity of the Si–O–Si asymmetric stretching mode, Si–O–Si symmetric stretching, and Si–O–Si symmetric bending mode at 1090 cm^−1^, 800 cm^−1^, and 482 cm^−1^ [27,28,29].

Lastly, the degradation behaviors of un-treated and PEG-treated BG specimens were tested in SBF solution, which was followed by the ISO-10993-14 protocol. A couple of BG specimens were used to obtained the average and standard deviation of weight loss. First, the pellets were weighed and then immersed in SBF at the ratio of 1 g to 10 mL. The solutions were discarded each day while the pellets were dried and weighed again. The whole assessment refreshed the SBF every day and was repeated 30 times.

## 3. Results

### 3.1. Phase Composition and Morphology

Figure 1 shows the XRD patterns of un-treated and PEG-treated powders. First, for the un-treated powder, the XRD pattern shows the absence of any crystalline peak except for a broad band between the diffraction angles of 20° and 40°. This result suggests that the structure of un-treated BG powder is amorphous. Next, for the 0.1 M, 0.3 M, and 0.5 M PEG-treated powders, the similar XRD patterns have been detected, which supports that the PEG-treated powders are amorphous. Therefore, the result indicates that all BG specimens were prepared successfully.

Figure 2 shows the SEM micrographs for observing the surface morphologies of un-treated and PEG-treated BG particles. Initially, it can be seen from Figure 2a that the surface morphology of un-treated BG particles are smooth and spherical with sizes ranging from 0.15 to 2.5 μm. Afterward, for the PEG-treated BG particles, the similar surface morphology of the smooth sphere has been observed in Figure 2b–d. In addition, the paritcle sizes of 0.1 M, 0.3 M, and 0.5 M PEG-treated BG particles range from 0.2 to 3.6 μm, 0.2 to 4.0 μm, and 0.2 to 4.0 μm, respectively. In short, both un-treated and PEG-treated BG particles have similar surface morphology. Thus, further details of the inner structure were examined by TEM.

For the inner structure, Figure 3 shows the TEM images of all BG particles. To begin with, it should be noted that the contrast of TEM images is contributed by two factors of sample thickness and crystal orientation. In this study, all specimens are amorphous (confirmed by the XRD results as shown in Figure 1). Therefore, the contrast is only influenced by the factor of sample thickness. The bright and dark contrast contributes to thicker and thinner regions, respectively. For the un-treated BG specimen shown in Figure 3a, only the particles with continuous contrast were found, which indicates that there is no thickness variation within the particles. Thus, it can be recognized as a solid sphere (Type I). Then, besides Type I particles, Figure 3b shows that two new morphologies—porous sphere (Type II) that contains multiple pores within a particle and hollow sphere (Type III) that contains a single pore within a particle—can be observed from the 0.1 M PEG-treated BG particles. In addition, similar to the 0.1 M PEG-treated particles, the 0.3 M and 0.5 M PEG-treated BG particles also exhibit three distinct morphologies of a solid sphere, a porous sphere, and a hollow sphere, as shown in Figure 3c,d. In summary, by combining both SEM and TEM images, un-treated particles exhibit only one morphology of a solid sphere, whereas the PEG-treated particles exhibit three distinct particles of solid, porous, and hollow structures.

Based on the statistical measurements from SEM and TEM images, the relative fractions of three distinct morphologies (solid, porous, and hollow) of all BG particles are shown in Figure 4. First, the un-treated BG particles have a single moprhology of a solid structure (the population of Type I is 100%). Second, unlike the un-treated BG case, three types of morphology have been found in all PEG-treated BG particles. For the 0.1 M PEG-treated specimen, the populations of Types I, II, and III are 22.4 ± 2.2%, 29.9 ± 3.2%, and 48.3 ± 0.9%, respectively. For the 0.3 M PEG-treated specimen, the populations of Type I, II, and III are 15.2 ± 1.2%, 31.9 ± 1.6%, and 52.3 ± 1.0%, respectively. For the 0.5 M PEG-treated specimen, the populations of Types I, II, and III are 11.8 ± 1.2, 26.8 ± 1.6, and 61.3 ± 1.3%, respectively. In addition, all p-values are lower than 0.05, which shows a distinct difference between un-doped and PEG-doped specimens. Furthermore, it can be seen from the graph that the population of solid particles (Type I) decreases with the increase of PEG concentration (100% for un-treated BG, and 22.4 ± 2.2%, 15.2 ± 1.2%, and 11.8 ± 1.2% for 0.1 M, 0.3 M, and 0.5 M PEG-treated BG), whereas Type III populations increase when the PEG concentration increases (0% for un-treated and 48.3 ± 0.9%, 52.3 ± 1.0%, and 61.3 ± 1.3% for 0.1 M, 0.3 M, and 0.5 M PEG-treated BG) respectively, which suggests that a higher concentratoin of PEG enhances the formation of hollow particles (Type III).

In addition, statistical analysis shows the average particle sizes of 1.05 ± 0.48, 1.29 ± 0.57, 1.54 ± 0.66, and 1.60 ± 0.69 μm for the un-treated BG powder, with 0.1 M, 0.3 M, and 0.5 M PEG-treated BG powders, respectively. It can be seen from the data that the particle size increases with the increase of PEG concentration. Additionally, the relationship between particle morphology and size is shown below. In the case of 0.1 M PEG-treated specimen, Types I, II, and III particles are 0.67 ± 0.09 μm, 0.87 ± 0.12 μm, and 1.30 ± 0.19 μm, respectively. As for 0.3 M PEG-treated specimen, Type I, II, and III particles are 0.80 ± 0.18 μm, 0.93 ± 0.14 μm, and 1.38 ± 0.45 μm, respectively. At last, for the 0.5 M PEG-treated specimen, Type I, II, and III particles are 0.83 ± 0.17, 1.13 ± 0.19, and 1.48 ± 0.22 μm, respectively. From these results, it is clear that the order of the particle size is a hollow particle (Type III) > porous particle (Type II) > solid particle (Type I), which indicates that particle morphology is related to particle size. Following Equation (1), the porosity values of the 0.1 M, 0.3 M, and 0.5 M PEG-treated particles can be computed as 45.8%, 68.4%, and 71.6%, respectively, which suggests that the higher PEG concentation leads the higher porosity. In brief, the size and porosity of the particle increases with the addition of PEG, and the moprhology can be related to its particle size.

### 3.2. Specific Surface Area and Bioactivity, In-Vitro Degradation Test

The BET measurements reveal the specific surface areas of the un-treated BG specimen. Furthermore, 0.1 M, 0.3 M, and 0.5 M PEG-treated BG specimens are 40.5 ± 0.1 m^2^/g, 56.5 ± 5.4 m^2^/g, 69.7 ± 1.8 m^2^/g, and 87.6 ± 6.5 m^2^/g, respectively. The results clearly show that higher concentration of the PEG additive will induce the higher specific surface area.

For the in vitro bioactivity tests, Figure 5 shows the SEM images of un-treated, 0.1 M, 0.3 M, and 0.5 M PEG-treated BG specimens after immersing in SBF for one day. Initially, the SEM images from the un-treated BG after SBF immersion reveals that some small crystallites (< 100 nm) are formed on the BG surfaces, as shown in Figure 5a. In addition, Figure 5b–d show similar results with small crystallites formed on the surfaces of PEG-treated BG particles after being immersed in SBF for one day. In order to identify the compositions and amounts of crystal phases, further investigations of XRD and FTIR are discussed below.

Figure 6 shows the XRD patterns of un-treated, 0.1 M, 0.3 M, and 0.5 M PEG-treated BG after immersing in SBF for one day. Compared to XRD patterns of as-prepared BG powders shown in Figure 1, diffraction peaks of (002) and (211) were observed in all BG powders, which correspond to the formation of hydroxyapatite (HA, JCPDF No.89-6495) after immersing in SBF solution. Furthermore, to determine the bioactivity of each specimen, the peak area analysis of HA is conducted to quantify the crystallinity (diffraction peak area). The order of the peak area corresponding to the (211) plane was 0.5 M PEG-treated BG (589 a.u.) > 0.3 M PEG-treated BG (408 a.u) > 0.1 M PEG-treated BG (384 a.u) > un-treated BG powders (220 a.u). The higher HA crystallinity (diffraction peak area) is, the higher the bioactivity value becomes. In addition, the XRD results demonstrate that 0.5 M PEG-treated BG powder gives the largest peaks at (211) as compared to the rest of BG specimens, which indicates higher crystallinity of HA. This corresponds to higher bioactivity.

Next, Figure 7 shows the FTIR spectra of in-vitro bioactivity of un-treated 0.1 M, 0.3 M, and 0.5 M PEG-treated BG powders. As shown in Figure 7a, before immersing in SBF, all specimens show the Si–O–Si bands at 1090 cm^−1^, 800 cm^−1^, and 482 cm^−1^ [28,29]. In contrast, after soaking in SBF for one day, all specimens show a new additional peak that appears at 566 cm^−1^ and is assigned to the P–O bending vibrations, as shown in Figure 7b. To determine the bioactivity of each specimen, the I_1_/I_2_ values were computed as 0.37, 0.83, 0.85, and 0.88 for un-treated powders, and 0.1 M, 0.3 M, and 0.5 M PEG-treated BG powders, respectively. Since the higher I_1_/I_2_ value indicates better bioactivity, it can be concluded that the PEG-treated BG specimens exhibit better bioactivity than the un-treated specimen.

For degradation behavior, Figure 8 shows the weight loss of un-treated, 0.1 M, 0.3 M, and 0.5 M PEG-treated BG pellets in SBF solution for 30 days. It can be seen from the graph that the weight loss of the un-treated BG sample reached ~6.5% after being immersed in SBF solution for 30 days. This results in an average weight loss of ~0.22% per day. With the additive of PEG, the values of total weight loss increase dramatically (20.4%, 27.4%, and 34.0% for the 0.1 M, 0.3 M, and 0.5 M PEG-treated BG powders), which results in the average degradation rates of the 0.1 M, 0.3 M, and 0.5 M PEG-treated BG powders being ~0.68%/day, ~0.91%/day, and ~1.13%/day, respectively. In summary, the PEG-treated BG powders show a higher degradation rate than the un-treated BG powder. In addition, the higher PEG concentration results in a higher degradation rate.

## 4. Discussion

First, based on previous studies [24], there are two main formation mechanisms, “one-particle-per-drop” and “gas-to-particle”, dominating the particle morphology during the SP process. Based on the SEM images shown in Figure 2, the average particle sizes of all spray pyrolyzed specimens are larger than 0.1 μm with the absence of nano-sized particles. This indicates that all droplets went through the SP process following the “one-particle-per-drop” formation mechanism rather than “gas-to-particle” conversion. Next, according to TEM images shown in Figure 3, three inner structures of solid (Type I), porous (Type II), and hollow (Type III) structures were found among all BG specimens. For the un-treated BG particles shown in Figure 3a, only solid spherical particles were observed. This is due to the typical “volume precipitation” mechanism of the SP process, which has been studied in our previous work [30]. In addition, for the PEG-treated BG particles (Figure 3b–d), two additional morphologies of porous and hollow structures can be observed. Furthermore, statistical measurements show that the increasing concentration of PEG will result in larger particle sizes, a higher portion of hollow particles, and higher porosity. Since the ultrasonic frequency and heat treatments during the SP process are identical for all productions, the variations of the particle size and inner structure are due to the PEG additive. To begin with, it is well known that the decomposition of PEG will form pores within the particle, while a higher amount of PEG forms a larger volume of pores. Hence, the particle formation mechanism of PEG-treated BG particles can be summarized as follows. First, droplets contained a smaller concentration of PEG after ultrasonic atomization. Solid particles were formed via the volume precipitation mechanism [30]. Next, with the increasing concentration of PEG, atomized droplets will contain more PEG and, thus, porous particles were formed [31]. At last, droplets contained a higher concentration of PEG. The precursor will precipitate on the surface while the PEG tends to stay in the core. Therefore, the hollow structure is formed once the temperature reached the decomposition temperature of PEG (450 °C) [20]. To note, the calcination temperature of BG is 550 °C [21], which allows the decomposition of PEG to complete.

Next, the correlation between morphology and porosity is discussed below. Figure 4 demostrates the relative fractions of solid (Type I), porous (Type II), and hollow (Type III) particles for the un-treated and PEG-treated BG particles. It can be seen from the graph that the population of Type II and Type III particles increases with a growing PEG concentration (combined ratios of Type II and Type III are 0%, 78.2 ± 4.1%, 84.2 ± 2.6%, and 88.1 ± 2.9% for un-treated, 0.1 M, 0.3 M, and 0.5 M PEG-treated particles, respectively). In addition, the computed porosity values are 0%, 45.8%, 68.4%, and 71.6%, which correspond to un-treated, 0.1 M, 0.3 M, and 0.5 M PEG-treated powders. By correlating the population of porous and hollow particles to porosity, it is clear that the porosity of BG powders are mainly contributed from Type II and Type III populations. This result supports that, by increasing the PEG concentration, more pores have been created and, thus, higher porosity is produced.

In addition, the relationships between porosity, the specific surface area, and bioactivity (I_1_/I_2_) are discussed in Figure 9. Initially, it can be seen from Figure 9a that the porosity values of 0%, 45.8%, 68.4%, and 71.6% correspond to specific surface areas of 40.5 ± 0.1, 56.5 ± 5.4, 69.7 ± 1.8, and 87.6 ± 6.5 m^2^/g, respectively. These results show that higher porosity values give a higher specific surface area. Moreover, Figure 9b demonstrates that BG specimens with a higher specific surface area will lead to a higher I_1_/I_2_ value (i.e., better bioactivity), which also agrees with our previous study [32].

Lastly, the correlations of PEG concentration, the porosity, and the degradation rate are discussed in Figure 10. To start with, Figure 10a shows that, when the PEG concentration increased, the porosity values increased as well (e.g. 45.8% for 0.1 M PEG-treated powder and 71.6% for 0.5 M PEG-treated powder). Then, with the increase of the porosity value, the degradation rate will also increase, as shown in Figure 10b. Both results agree with Zhang et al. [9,33]. In summary, a BG specimen prepared with higher PEG concentration will lead to porous and hollow structures with higher porosity, which then contributes to faster degradation.

## 5. Conclusions

In this study, both un-treated and PEG-treated BG powders were successfully synthesized using spray pyrolysis. The surface morphologies and inner structures were examined by SEM and TEM, which revealed three morphologies of solid, porous, and hollow spheres. With the increase of PEG concentration, higher populations of porous and hollow particles were observed and the powders exhibited a higher specific surface area and higher bioactivity. In addition, the assessments of degradation behavior suggested that the porous and hollow structures created in PEG-treated BG specimens can speed up the degradation rate.

## Figures and Tables

**Figure 1 materials-12-03703-f001:**
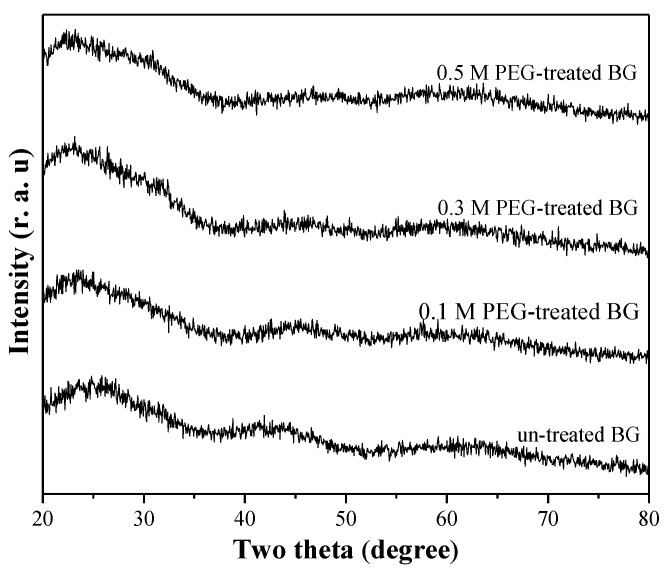
XRD patterns of un-treated BG powder and 0.1 M, 0.3 M, and 0.5 M PEG-treated BG powders.

**Figure 2 materials-12-03703-f002:**
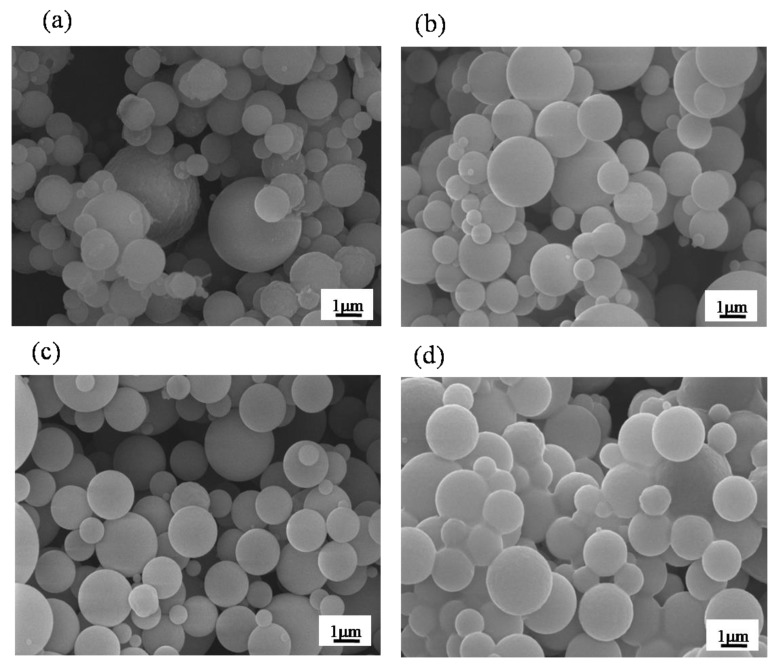
SEM images of (**a**) un-treated BG particles and (**b**) 0.1 M, (**c**) 0.3 M, and (**d**) 0.5 M PEG-treated BG particles.

**Figure 3 materials-12-03703-f003:**
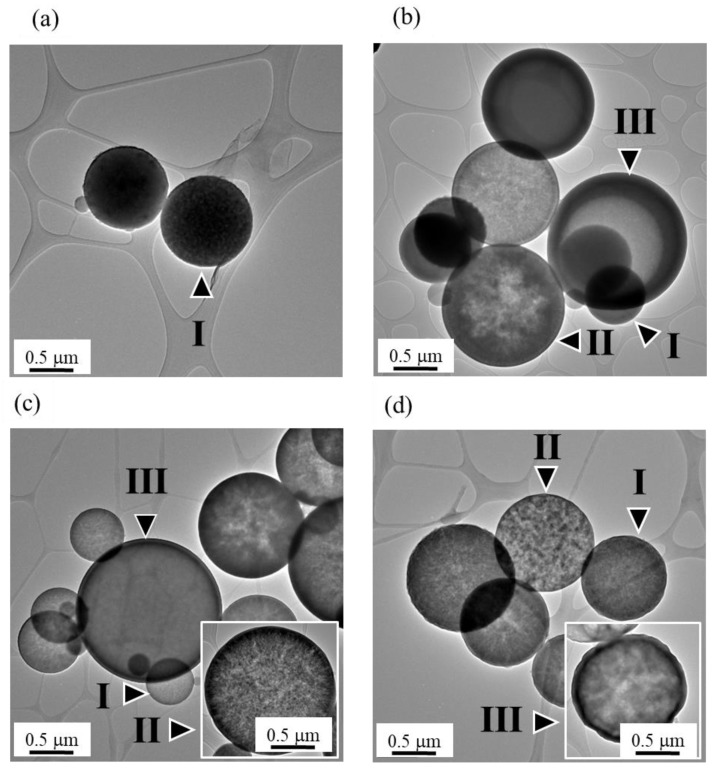
TEM images of (**a**) un-treated BG particles and (**b**) 0.1 M, (**c**) 0.3 M, and (**d**) 0.5 M PEG-treated BG particles. (Type I, Type II, and Type III represents solid, porous, and hollow particles).

**Figure 4 materials-12-03703-f004:**
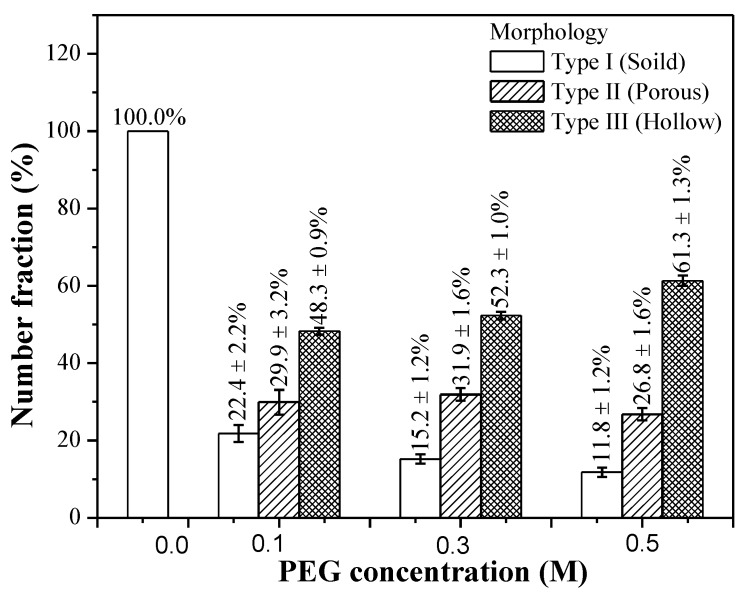
Relative fractions of un-treated 0.1 M, 0.3 M, and 0.5 M PEG-treated BG particles with three distinct morphological types of solid, porous, and hollow structures.

**Figure 5 materials-12-03703-f005:**
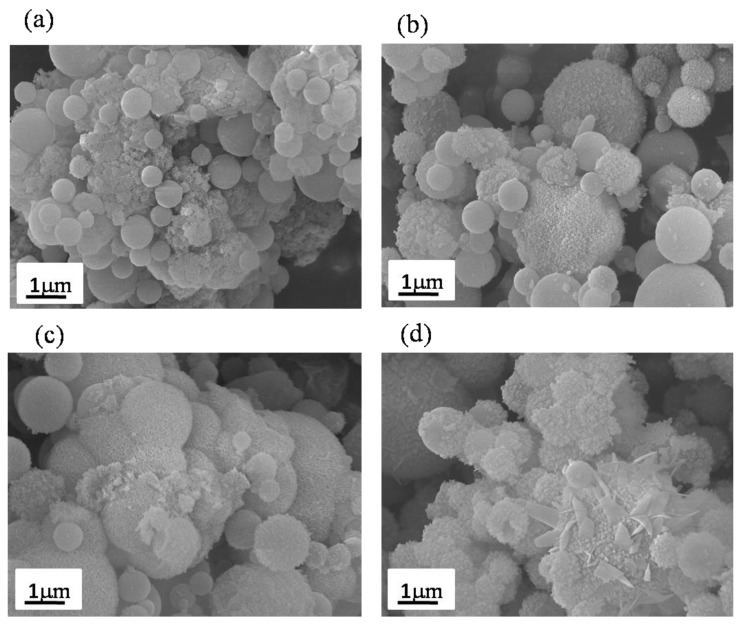
SEM images of (**a**) un-treated BG particles and (**b**) 0.1 M, (**c**) 0.3 M, and (**d**) 0.5 M PEG- treated BG powders after being immersed in SBF for one day.

**Figure 6 materials-12-03703-f006:**
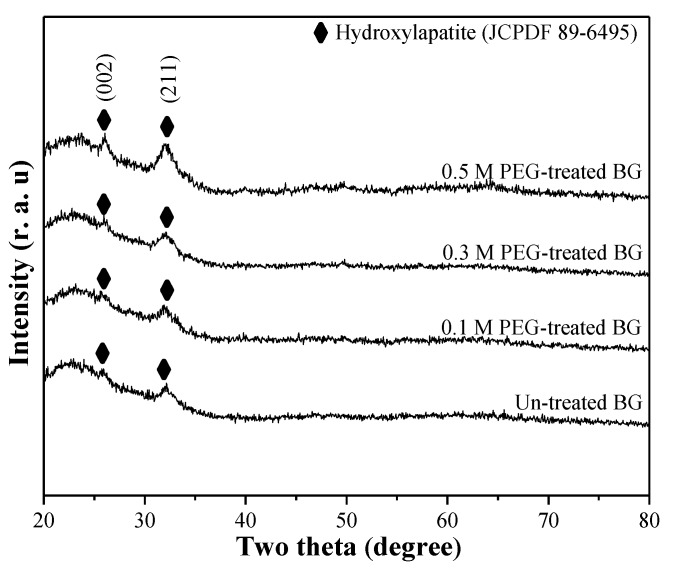
XRD patterns of un-treated BG powder and 0.1 M, 0.3 M, and 0.5 M PEG-treated BG powders after being immersed in SBF for one day.

**Figure 7 materials-12-03703-f007:**
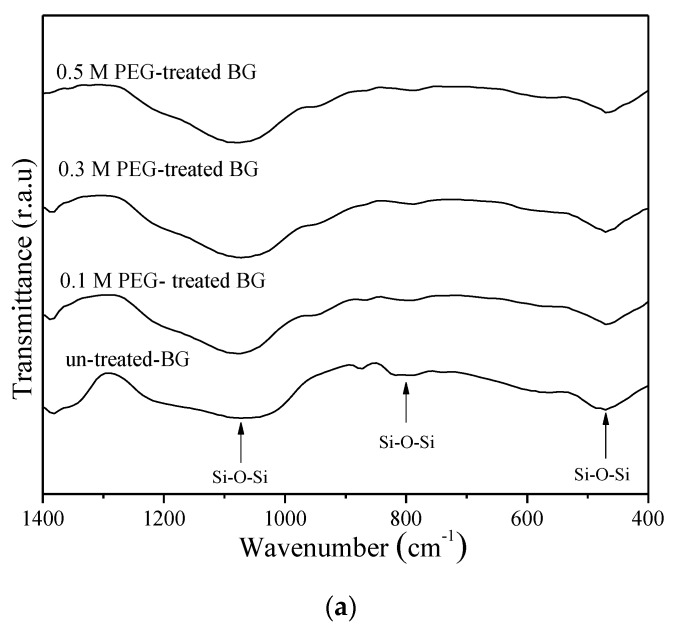
FTIR spectra of un-treated, 0.1 M, 0.3 M, and 0.5 M PEG-treated BG powders (**a**) before and (**b**) after being immersed in SBF for one day.

**Figure 8 materials-12-03703-f008:**
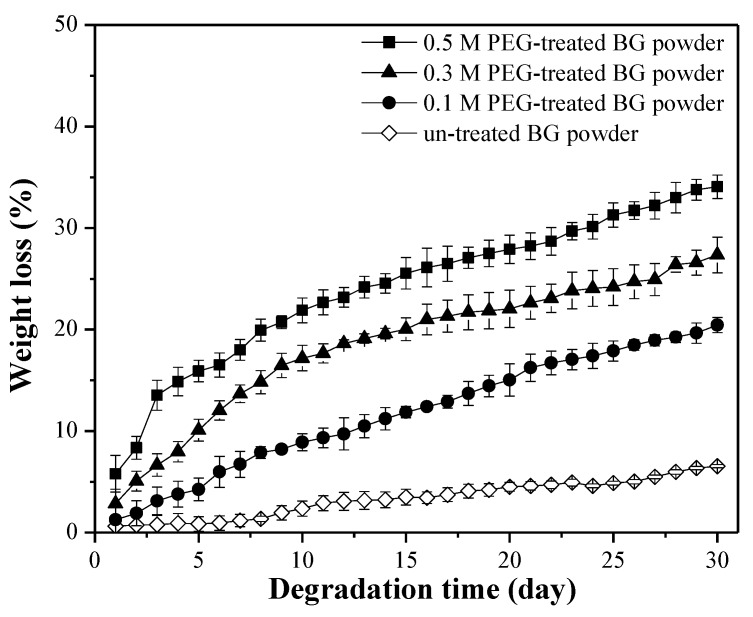
Weight loss of un-treated, 0.1 M, 0.3 M, and 0.5 M PEG-treated BG powders in SBF solution as a function of time.

**Figure 9 materials-12-03703-f009:**
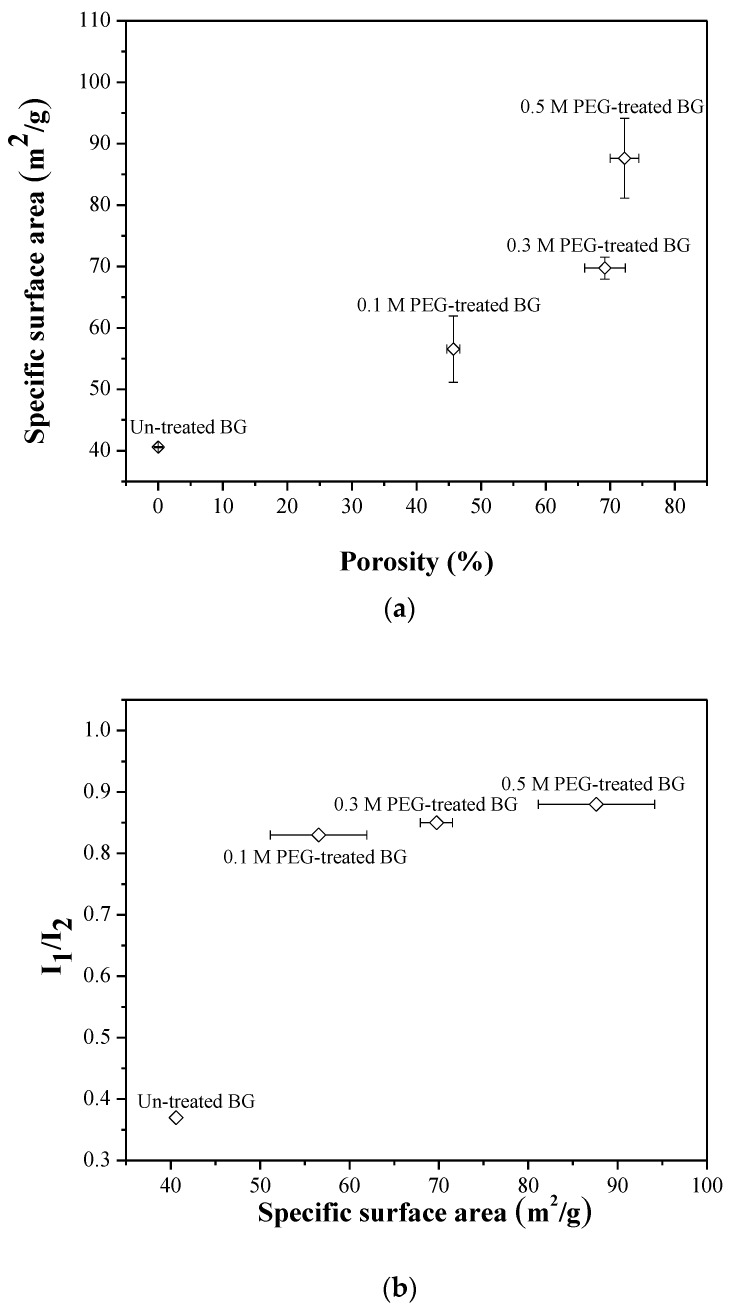
Correlations of (**a**) specific surface area to porosity and (**b**) bioactivity to a specific surface area.

**Figure 10 materials-12-03703-f010:**
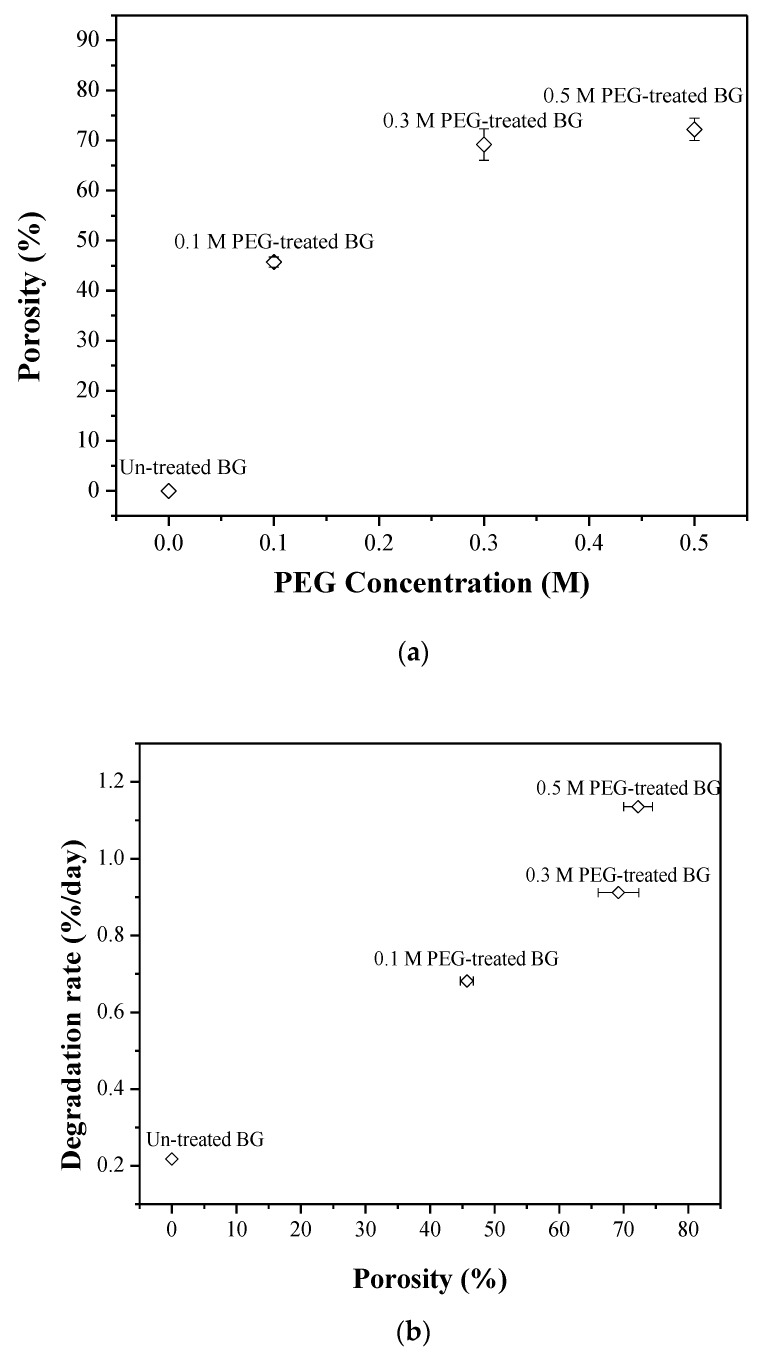
Correlations of (**a**) porosity to PEG concentration and (**b**) degradation rate to porosity.

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
