# Peer review of "Correlation of Morphology and In-Vitro Degradation Behavior of Spray Pyrolyzed Bioactive Glasses"

_materials, 2019, doi:10.3390/ma12223703_

Round 1

Reviewer 1 Report

The editing of the paper is in an disastrous state.

Hundreds of words are not separated by blanks. Examples page 1, line 24:  … totheir …., line 28: … onmigration …. Citations should be seperated from the texts by a blank: Example: Page 1, line 25: bones[1,2] After points commas, semicolon blanks should separate the text from the punctuation marks. Blanks should also separate numbers from units.

Overall the disastrous state of the editing makes it very inconvenient to read the text.

Therefore, a readable text should be prepared and then he might be reviewed. But not in the current state.

Author Response

We would like to apologize for the wrong format during exporting. We have corrected the manuscript for further arrangement.

Reviewer 2 Report

This is an interesting contribution to bioglass/biomaterials science. I would like to recommend publication.

I would also like to give some suggestions to the authors for further improvement:

Proof-read the text, some blank spaces are missing between the words.

Introduction is a dry and should be expanded. You could cite for example:

Electrophoretic deposition of spray-dried Sr-containing mesoporous bioactive glass spheres on glass–ceramic scaffolds for bone tissue regeneration. Journal of Materials Science 2017;52:9103-9114

Bioactive glasses: from parent 45S5 composition to scaffold-assisted tissue-healing therapies. Journal of Functional Biomaterials 2018;9:24

Figure 4: statistical analysis is missing: +/- SD? p-value?

4. No interpolation of data (modelling) is presented in Figures 7 and 8. Please improve this part.

Author Response

This is an interesting contribution to bioglass/biomaterials science. I would like to recommend publication.

I would also like to give some suggestions to the authors for further improvement:

Proof-read the text, some blank spaces are missing between the words.

We are sorry for this mistake. All missing blank spaces have been inserted.

Introduction is a dry and should be expanded. You could cite for example:

Electrophoretic deposition of spray-dried Sr-containing mesoporous bioactive glass spheres on glass–ceramic scaffolds for bone tissue regeneration. Journal of Materials Science 2017;52:9103-9114

Bioactive glasses: from parent 45S5 composition to scaffold-assisted tissue-healing therapies. Journal of Functional Biomaterials 2018;9

Many thanks for the useful suggestions. All suggested papers have been cited and the introduction is expanded.

Figure 4: statistical analysis is missing: +/- SD? p-value?

We would like to thank you for the reminding. The details of standard deviations and p-values have been added. Initially, for the standard deviations, for the 0.1M PEG-treated specimen, the populations of Type I, II, and III are 22.4±2.2%, 29.9±3.2%, and 48.3±0.9%, respectively; for the 0.3 M PEG-treated specimen, the populations of Type I, II, and III are 15.2±1.2%, 31.9±1.6%, and 52.3±1.0%, respectively; for the 0.5 M PEG-treated specimen, the populations of Type I, II, and III are 11.8±1.2%, 26.8±1.6%, and 61.3±1.3%, respectively. Secondly, the p-values of populations for Type I, II, and III are 0.0001392, 0.0016602, and 0.0000579 for the 0.1 M PEG-treated specimen; 0.0000339, 0.0004331, and 0.0000026 for the 0.3 M PEG-treated specimen; 0.0000286,      0.0005969, and 0.0000701 for the 0.5 M PEG-treated specimen, which indicates that all p-values are less than 0.05.

No interpolation of data (modelling) is presented in Figures 7 and 8. Please improve this part.

We have added the interpretation in the figures. Thank you very much.

Reviewer 3 Report

Authors have demonstrated the effect of polyethylene glycol (PEG) on the morphology and in vitro degradation behavior of PEG-treated bioactive glass (BG) particles. This work is interesting and provides new insight for researchers for hard tissue regeneration. The concept and design of the manuscript is good. The manuscript is well written and characterized. However, this manuscript can be accepted after the following minor revision.

Authors should also provide the SEM/XRD analysis of SBF-treated BG samples for their morphological/structural changes in both cases un-treated and PEG-treated BGs, in response to non-porous and porous particles.

Author Response

Authors have demonstrated the effect of polyethylene glycol (PEG) on the morphology and in vitro degradation behavior of PEG-treated bioactive glass (BG) particles. This work is interesting and provides new insight for researchers for hard tissue regeneration. The concept and design of the manuscript is good. The manuscript is well written and characterized. However, this manuscript can be accepted after the following minor revision.

Authors should also provide the SEM/XRD analysis of SBF-treated BG samples for their morphological/structural changes in both cases un-treated and PEG-treated BGs, in response to non-porous and porous particles.

We would like to appreciate the comments suggested by the reviewer. We have performed the SEM and XRD measurements (Figures 5 and 6) after immersed in SBF for 1 day. Both SEM and XRD results suggest that PEG-treated BG powders exhibit the better bioactivity than that of the un-treated BG powder.

Figure 5. SEM images of (a) un-treated BG particles and (b) 0.1M, (c) 0.3M and (d) 0.5M PEG- treated BG powders after immersed in SBF for 1 day.

Figure 6. XRD patterns of un-treated BG powder and 0.1M, 0.3M and 0.5M PEG-treated BG powders after immersed in SBF for 1 day.

Round 2

Reviewer 1 Report

Comment on F. F. Bakare et al.: Correlation of morphology and in-vitro degradation behaviour of spray pyrolyzed bioactive glasses version II

Generally

The English style of the chapters “Introduction” as well as “Materials and methods” has been greatly improved. The style, grammar and orthography of the following has still to be greatly improved.

Comments to specific points:

Page 2, line 58: If degradation is discussed as corrosion process (which it is) there is numerous literature on correlation between surface area and degradation rates: e.g.: Buckwalter, C. Q., Pederson, L. R. & McVay, G. L. The effects of surface area to solution volume ratio and surface roughness on glass leaching. Non-Cryst. Sol. 1982, 49, 397-412. Additionally the relation between porosity and surface area is also well established in textbooks of materials science or on porous materials. Page 2, lines 92 and following. Particle size analysis by measuring circles in TEM or SEM is possible but not the best method. Has it been tried by Fraunhofer diffraction? The results of it (page 4): a size distribution would be a better information. Page 3, line 117: degradation rate: how is it measured? On pge 10 values are reported in %. Possibly %/d was meant. On the other hand, degradation/corrosion rates are usually reported in mass loss per time unit and per unit of surface area (g×d-1×m-2). Figure 1. A closer view reveals three broad peaks with a shifted centre: what does it mean? Page 8, line 214. Is it possible to quantify crystallinity? 7: Is there any discussion of band assignment? Can the authors cite literature concerning band assignment? Usually the absorption bands between 1300 and 800 cm-1 are attributed different silicate groups, look in: King, P. L., McMillan, P. F. und Moore, G. M. Infrared spectroscopy of silicate glasses with application to natural systems. [Buchverf.] P. L. King (ed.). Molecules to planets: Infrared spectroscopy in geochemistry, exploration geochemistry and remoete sensing. Short course. Quebec, Canada : Mineralogical Association of Canada, 2004, S. 93-134. Figure 8: How do the authors evaluate accuracy, They have shown error bars (that’s good), but how are they calculated? PEG: How does it contribute to microstructure formation. Is it dissolved on a molecular basis, does it form structure directing units (micelles?). I I evaporated during heating? Does the solvent in the droplets evaporate and the PEG decomposes leaving hollow spheres behing? Evaluation in Figures 9 and 10. There are 4 pictures, of minor importance. Possibly a correlation between surface area and degradation rate would back the conclusion.

Author Response

Point 1: Page 2, line 58: If degradation is discussed as corrosion process (which it is) there are numerous literatures on correlation between surface area and degradation rates: e.g.: Buckwalter, C. Q., Pederson, L. R. &McVay, G. L. The effects of surface area to solution volume ratio and surface roughness on glass leaching. Non-Cryst. Sol. 1982, 49, 397-412. Additionally, the relation between porosity and surface area is also well established in textbooks of materials science or on porous materials.

Response 1:

Many thanks for the useful suggestions. The suggested paper has been cited, and the corresponding texts have been modified.

Point 2: Page 2, lines 92 and following. Particle size analysis by measuring circles in TEM or SEM is possible but not the best method. Has it been tried by Fraunhofer diffraction? The results of it (page 4): a size distribution would be better information.

Response 2:

Thank you very much for the constructive comments and for reminding us to use “Franuhofer diffraction” to analysis the particle size. However, we did not use Franuhofer diffraction to recalculate particle size because of the following two reasons. First, we are unable to access the instrument in this short period of time. Second, the method of Franuhofer diffraction cannot provide the correlation between morphology and particle size, which was discussed in this study. 

Point 3: Page 3, line 117: degradation rate: how is it measured? On page 10 values are reported in %. Possibly %/d was meant. On the other hand, degradation/corrosion rates are usually reported in mass loss per time unit and per unit of surface area (g×d-1×m-2).

Response 3:

The (average) degradation rate is calculated using the total weight loss of 30 days divided by 30 days. We would like to apologize for missing the unit for degradation rate, which is mass loss per time unit (%/day). We have corrected the unit in the manuscript.

Point 4: Figure 7. A closer view reveals three broad peaks with a shifted centre: what does it mean? Page 8, line 214. Is it possible to quantify crystallinity?

Response 4: 

We would like to thank you for the reminding. Indeed, the crystallinity of BG powders can be quantified. The details of crystallinity values have been added. The crystallinity (diffraction peak area) was analyzed from peak area of HA. The order of peak area corresponding to the (211) plane. The order of peak area corresponding to the (211) plane was 0.5 M PEG-treated BG (589 a.u.) > 0.3 M PEG-treated BG (408 a.u) > 0.1 M PEG-treated BG (384 a.u) > un-treated BG powders (220 a.u). Since the higher HA crystallinity (diffraction peak area), value indicates the higher the bioactivity.

Point 5: Page 7: Is there any discussion of band assignment? Can the authors cite literature concerning band assignment? Usually the absorption bands between 1300 and 800 cm-1 are attributed different silicate groups, look in: King, P. L., McMillan, P. F. und Moore, G. M. Infrared spectroscopy of silicate glasses with application to natural systems. [Buchverf.] P. L. King (ed.). Molecules to planets: Infrared spectroscopy in geochemistry, exploration geochemistry and remote sensing. Short course, Quebec, Canada: Mineralogical Association of Canada, 2004, S. 93-134.

Response 5:

Many thanks for the useful suggestions. The suggested paper has been cited, and the band assignment is discussed.

Point 6: Figure 8: How do the authors evaluate accuracy, they have shown error bars (that’s good), but how are they calculated?

Response 6:

Thank you very much for the question. In Figure 8, the accuracy of weight loss was determined by the average values and standard deviations, which were obtained by the couple of BG specimens. The description has been added in the experimental procedure.

Point 7: PEG: How does it contribute to microstructure formation? Is it dissolved on a molecular basis, does it form structure directing units (micelles?). Evaporated during heating? Does the solvent in the droplets evaporate and the PEG decomposes leaving hollow spheres behind?

Response 7:

PEG is an additive which is commonly used as a pore forming agent (Lei, B., J. Am. Ceram. Soc.  93: 32-35 (2010) and Lei, B., J. Sol. Gel. Sci. Technol., 58: 3 656–63 (2011)). It is a hydrophilic polymer and tends to stay dissolved in the solvent, thus there is low possibility for PEG to form micelles. However, instead of self-assembly process, PEG tends to aggregate in the core and the inorganic precursor is forced to go to the surface. Then, during the calcination stage, PEG (450 oC) was completely decomposed and resulting in pore (porous and hollow) particles are formed. To note, the calcination temperature of BG is 550 oC which allows the decomposition of PEG to be complete. 

Point 8: Evaluation in Figures 9 and 10. There are 4 pictures, of minor importance. Possibly a correlation between surface area and degradation rate would back the conclusion.

Response 8:

Many thanks for your suggestion. Thus figures explained the main purpose of this manuscript. Therefore, we prefer to leave the Figures 9 and 10 in the manuscript.
